# Development of a Crystal Digital RT-PCR for the Detection of Atypical Porcine Pestivirus

**DOI:** 10.3390/vetsci10050330

**Published:** 2023-05-04

**Authors:** Huixin Liu, Kaichuang Shi, Shuping Feng, Yanwen Yin, Feng Long, Hongbin Si

**Affiliations:** 1College of Animal Science and Technology, Guangxi University, Nanning 530005, China; 2Guangxi Center for Animal Disease Control and Prevention, Nanning 530001, China

**Keywords:** atypical porcine pestivirus (APPV), crystal digital RT-PCR (cdRT-PCR), real-time quantitative RT-PCR (qRT-PCR), detection method

## Abstract

**Simple Summary:**

Atypical porcine pestivirus (APPV), a newly discovered virus, is associated with the type A-II congenital tremor (CT) in neonatal piglets. Here, the specific primers and probe were designed, the reaction conditions were optimized, and a crystal digital RT-PCR (cdRT-PCR) and real-time quantitative RT-PCR (qRT-PCR) for the accurate detection of APPV were successfully established. Both assays were demonstrated to be highly sensitive, strong specific, and excellently repeatable. The sensitivity of the cdRT-PCR was 100 times higher than that of the qRT-PCR. The 60 clinical tissue samples were tested for APPV, showing 23.33% positivity by the qRT-PCR and 25% positivity by the cdRT-PCR, with a coincidence rate of 98.33%. The results illustrate that both established methods are useful tools for detection of APPV.

**Abstract:**

Atypical porcine pestivirus (APPV), a newly discovered virus, is associated with the type A-II congenital tremor (CT) in neonatal piglets. APPV distributes throughout the world and causes certain economic losses to the swine industry. The specific primers and probe were designed targeting the 5′ untranslated region (UTR) of APPV to amplify a 90 bp fragment, and the recombinant standard plasmid was constructed. After optimizing the concentrations of primers and probe, annealing temperature, and reaction cycles, a crystal digital RT-PCR (cdRT-PCR) and real-time quantitative RT-PCR (qRT-PCR) were successfully established. The results showed that the standard curves of the qRT-PCR and the cdRT-PCR had R^2^ values of 0.999 and 0.9998, respectively. Both methods could specifically detect APPV, and no amplification signal was obtained from other swine viruses. The limit of detection (LOD) of the cdRT-PCR was 0.1 copies/µL, and that of the qRT-PCR was 10 copies/µL. The intra-assay and inter-assay coefficients of variation of repeatability and reproducibility were less than 0.90% for the qRT-PCR and less than 5.27% for the cdRT-PCR. The 60 clinical tissue samples were analyzed using both methods, and the positivity rates of APPV were 23.33% by the qRT-PCR and 25% by the cdRT-PCR, with a coincidence rate of 98.33%. The results indicated that the cdRT-PCR and the qRT-PCR developed here are highly specific, sensitive methods for the rapid and accurate detection of APPV.

## 1. Introduction

Atypical porcine pestivirus (APPV), an enveloped, single-stranded RNA virus with a genome of about 11 kb, belongs to the genus *Pestivirus* of the family *Flaviviridae* [1]. Classical swine fever virus (CSFV), border disease virus (BDV), bovine viral diarrhea virus 1 (BVDV-1), and BVDV-2 also belong to the same family. APPV was first found in domestic pigs in the United States of America (USA) in 2015 [2], and then identified to be associated with the clinical piglets with congenital tremor (CT) in 2016 [3]. Subsequently, APPV was reported in many countries, such as China, Germany, Italy, Great Britain, Austria, Switzerland, Spain, Hungary, Brazil, Danmark, and Japan [4,5,6,7,8,9,10,11]. Guangdong was the first Chinese province to report APPV in 2016 [4], followed by its discovery in Guangxi, Yunnan, Guizhou, Anhui, Jiangxi, Henan, and other provinces [12,13,14,15,16,17]. The infected piglets are clinically characterized by the type A-II congenital tremor, and some occasionally display splay legs [18,19], while adult pigs may present persistent infection but intermittently shed virus [20,21,22]. APPV can infect domestic pigs and wild boars [23,24]. High APPV loads were founded in the semen, serum, cerebellum, cerebrum, spinal cord, brainstem, kidney, lymph nodes, thymus, heart, spleen, feces, and nasal secretions [20,25,26], meaning that a nasal swab, fecal swab, and different tissue samples are suitable for detection of APPV. APPV can be horizontally transmitted via the oronasal pathway, as well as vertically transmitted via transplacental infection [3,27]. The infected pig herds showed a decreased survival rate and growth performance, with severe economic damages to pig production [20,28]. Furthermore, since the disease shows similar manifestations with classical swine fever (CSF), pseudorabies (PR), and some other diseases, which are still epidemic in some pig herds in many countries [29,30], it is very hard to accurately differentiate and diagnose them only depending on the clinical signs and pathological damages. Therefore, a highly sensitive and specific method for accurate detection of APPV is necessary in order to facilitate efficient surveillance, prevention and control of this disease.

Currently, some assays for the detection of APPV nucleic acid, including reverse transcribed (RT) polymerase chain reaction (PCR), multiplex RT-PCR, real-time quantitative RT-PCR (qRT-PCR), and multiplex qRT-PCR, have been reported [31,32,33,34]. The digital PCR (dPCR), a new molecular technique with extremely high sensitivity and precision for low template concentrations, can quantify nucleic acids by correcting the Poisson distribution to obtain the absolute copy number of the target gene without using standard curves [35,36]. The dPCR/dRT-PCR shows low sensitivity to PCR inhibitors, high accuracy, and repeatability [37]. So far, two types of digital PCR, the droplet digital PCR (ddPCR) and crystal digital PCR (cdPCR), have been reported [35,36]. Recently, a ddRT-PCR for test of APPV was reported [38]. However, no cdRT-PCR has ever been reported for the detection of APPV until now. APPV shows a high degree of genetic diversity among different strains from different countries or even from the same country [4,5,12,15,20,39], with as high as 21% genetic distance [5]. APPV contains a complete genome of about 11 kb, in which the 5′ untranslated region (UTR), N^pro^, E2, NS3, and NS5A genes are relatively conserved [12,14,15,20,39], and are usually used as target genes for designing the specific primers and probes to detect APPV. Here, a cdRT-PCR, using the Naica^TM^ System (Stilla Technologies, Villejuif, France) [36], was developed for detecting the 5′ UTR of APPV, and then applied to analyze the clinical samples from piglets with congenital tremor (CT).

## 2. Materials and Methods

### 2.1. Vaccine Strains and Clinical Samples

The vaccine strains, including CSFV (C strain), porcine circovirus type 2 (PCV2, ZJ/C strain), porcine reproductive and respiratory syndrome virus (PRRSV, JXA1-R strain), swine influenza virus (SIV, TJ strain), pseudorabies virus (PRV, Bartha-K61 strain), and foot-and-mouth disease virus (FMDV, O/Mya98/XJ/2010 strain), came from a commercial company (Huapai, Chengdu, China). The clinical tissue samples, which were confirmed to be positive for APPV, African swine fever virus (ASFV), BVDV-1, and BVDV-2, were provided by our laboratory. The vaccines and tissue samples were stored in freezing conditions (−80 °C) until use.

The 60 clinical tissue samples were collected from 15 pig farms in Guangxi province, southern China, during January 2020 and December 2021. The tissues came from 60 newborn piglets with CT and splay legs. The tissues, including liver, spleen, tonsils, and lymph nodes from each piglet, were stored in freezing conditions (−80 °C) until use.

### 2.2. Primers and Probe

According to the APPV genomic sequence (GenBank accession No. KY624591), the specific primers and TaqMan probe targeting the 5′ UTR (Table 1) were designed, and then used to develop the cdRT-PCR and the qRT-PCR.

### 2.3. RNA Extraction and Reverse Transcription

The RNA was extracted from a volume of 200 μL of viral solution or tissue homogenized supernatants from clinical samples using the Ex-DNA/RNA Extraction Kit (Tilong, China). The cDNA was transcribed by the PrimeScript^TM^ II 1st Strand cDNA Synthesis Kit (TaKaRa, Dalian, China).

### 2.4. Preparation of Standard Plasmid

A 90 bp fragment was amplified by PCR from the cDNA of APPV using APPV-F/R primers. The amplified fragment was purified, and then used to construct the recombinant standard plasmid as described by Liu et al. [29]. The recombinant plasmid was named p-APPV, and then used as standard plasmid. The values at 260 nm and 280 nm were measured, and the plasmid concentration was calculated as described by Liu et al. [29].

### 2.5. Optimization of the Reaction Conditions

The standard plasmid p-APPV of 10^4^ copies/µL (final reaction concentration: 10^3^ copies/µL) was used for qRT-PCR using the designed primers and probe with a 25 µL reaction system. The matrix method was used to determine the optimal primer and probe concentrations (25 pmol/µL of 0.1, 0.2, 0.3, 0.4, 0.5, and 0.6 µL). Amplification conditions: 95 °C for 15 s, 40 cycles at 95 °C for 5 s, and 56–60 °C for 5 s. All the experiments were performed using the Q6 qPCR system (ABI, Carlsbad, CA, USA).

The p-APPV plasmid with 10^4^ copies/µL (final reaction concentration: 10^3^ copies/µL) was used for the cdRT-PCR using the designed specific primers and probe with a 25 µL reaction system. The different concentrations of primers (from 600 to 1400 nM) and probe (from 200 to 400 nM), and the different annealing temperatures (56–61 °C) were used to optimize the reaction conditions. The amplification conditions were as follows: 95 °C for 15 s, 40 cycles at 95 °C for 5 s, and 56–60 °C for 5 s. The absolute concentration of the target nucleic acid was determined using Poisson statistics. All experiments were performed using the Naica^TM^ cdPCR system (Stilla Technologies^TM^, Villejuif, France).

### 2.6. Generation of the Standard Curves

The p-APPV plasmid was diluted from 10^5^ to 10^0^ copies/µL as the template, and the qRT-PCR was performed under the optimal reaction conditions. The Ct values were obtained, and the standard curve was generated.

The p-APPV plasmid was diluted from 10^4^ to 10^−1^ copies/µL as the template, and the cdRT-PCR was performed under the optimal reaction conditions. The number of positive droplets was automatically determined by the Naica^TM^ Prism 3 program (Stilla Technologies, Villejuif, France), and the standard curve was generated.

### 2.7. Specificity Analysis

The cDNA of APPV, CSFV, SIV, PRRSV, FMDV, BVDV-1, and BVDV-2, and the DNA of ASFV, PRV and PCV2 were used as templates, with RNase-free distilled water as a negative control, to evaluate the specificity of both established methods.

### 2.8. Sensitivity Analysis

The p-APPV plasmid was diluted from 10^6^ to 10^−1^ copies/µL for evaluation of the sensitivity of both developed methods.

### 2.9. Repeatability Analysis

The p-APPV plasmid was diluted from 10^4^ to 10^2^ copies/µL (the final concentrations in the reaction system ranged from 10^3^ to 10^1^ copies/μL), and then used to analyze the repeatability of both methods. The intra-assay and inter-assay coefficients of variation (CVs) were calculated in triplicate and on three different days, respectively.

### 2.10. Evaluation of the Clinical Samples

Both developed methods were applied to test the 60 clinical tissue samples. The coincidence rate and Kappa value between these two methods were analyzed.

## 3. Results

### 3.1. Acquirement of the Standard Plasmid

The cDNA of APPV was used to amplify the targeted fragment of 90 bp by PCR. The amplified product was purified to construct the recombinant plasmid. The positive colonies were cultured, and the recombinant plasmid was extracted and named p-APPV. The plasmid construct was verified by sequencing. The original concentration of the plasmid was calculated as 1.87 × 10^10^ copies/µL. The plasmid was diluted to 10^8^ copies/µL and stored at −80 °C.

### 3.2. Optimization of the Reaction Conditions

After optimization, the optimal primer and probe concentrations, annealing temperature, and amplification cycles were acquired for the qRT-PCR. The reaction system in a total volume of 25 µL is shown in Table 2. The amplification conditions were as follows: 95 °C for 15 s, then 40 cycles at 95 °C for 5 s and 59 °C for 34 s. The criterion for determining a positive sample was a Ct value ≤35.

After optimization, the optimal primer and probe concentrations, annealing temperature, and amplification cycles were acquired for the cdRT-PCR (Figure 1). The 25 µL reaction system is shown in Table 2. The amplification conditions were as follows: 95 °C for 15 s, then 40 cycles at 95 °C for 5 s and 59 °C for 34 s. Data were collected by crystal microdroplet array using the Nacia^TM^ Prism 3 instrument (Stilla Technologies, Villejuif, France).

### 3.3. Generation of Standard Curve

The p-APPV plasmids were diluted from 10^6^ to 10^0^ copies/μL, and then amplified by the established qRT-PCR. The standard curve was automatically generated, showing good linearity between the initial concentrations and the Ct values, with an equation slope of −3.716, a correlation coefficient (R^2^) of 0.999, and an amplification efficiency (E) of 103.28% (Figure 2A).

The p-APPV plasmids were diluted from 10^4^ to 10^−1^ copies/μL, and then amplified by the developed cdRT-PCR. The standard curve was generated; the equation slope was 1.03, and the R^2^ was 0.9998 (Figure 2B), indicating good linearity between the number of initial templates and positive droplets.

### 3.4. Analysis on the Specificity

The specificity of both developed methods was analyzed using the DNA/cDNA of APPV, CSFV, PRRSV, FMDV, SIV, BVDV-1, BVDV-2, ASFV, PRV, and PCV2. The fluorescence signals were obtained only from APPV, but not from other viruses (Figure 3), indicating high specificity of both methods for the detection of APPV.

### 3.5. Analysis on the Sensitivity

The sensitivity of both developed methods was evaluated using the p-APPV plasmid ranging from 10^6^ to 10^−1^ copies/μL. The results demonstrated that the detection limit of the qRT-PCR was 10 copies/μL, while the detection limit of the cdRT-PCR was 0.1 copies/μL, indicating that the latter was 100 times more sensitive than the former (Figure 4).

### 3.6. Analysis on the Repeatability

The repeatability of both developed methods was evaluated using the p-APPV plasmid with three concentrations of 10^3^, 10^2^, and 10^1^ copies/μL (final concentrations). The results demonstrated that the intra-assay CVs for repeatability ranged from 1.99% to 5.27% of the cdRT-PCR and from 0.60% to 0.90% of the qRT-PCR, while the inter-assay CVs for reproducibility ranged from 0.58% to 2.11% of the cdRT-PCR and from 0.34% to 0.79% of the qRT-PCR (Table 3), showing the excellent repeatability of both developed methods.

### 3.7. Evaluation of the Field Samples

The 60 tissue samples were evaluated using both developed methods. The positivity rates of APPV were 23.33% (14/60) by qRT-PCR and 25.00% (15/60) by cdRT-PCR. The coincidence rate was 98.33%, and the Kappa value was 0.955, demonstrating excellent agreement between both methods (Table 4).

## 4. Discussion

APPV is a newly discovered swine pathogen, which has been reported in many countries [4,5,6,7,8,9,10,11]. Some provinces in China have reported this disease since it was discovered in the USA in 2015 [12,13,14,15,16,17]. The infected newborn piglets show CT and, occasionally, splay legs [18,19]. The mortality rate usually ranges from 0.6% to 30% [20], but might be as extreme as 100% if the piglets do not receive colostrum because of body tremors [40]. The infected pigs might show persistence and no obvious manifestation, but can shed virus intermittently [20,21,22], while the infected sows might transmit APPV through the placenta [3,27]. As a result, it is hard to diagnose this disease depending only on clinical observations. Therefore, it is vital to establish a specific and accurate method to test this pathogen, as well as evaluate the damage of the disease so as to implement credible measures for prevention and control of this disease. The qRT-PCR and cdRT-PCR are rapid, accurate methods for the detection of APPV and diagnosis of this disease.

The qPCR is a rapid, specific, sensitive, and high-throughput technology, which depends on the Ct values and the calibration curves [41]. The dPCR obtains the absolute quantification by calculating through Poisson statistics independent of calibration curves, illustrating the advantages of increased precision, lower susceptibility to PCR inhibitors, greater accuracy for low concentrations, and higher reliability and repeatability compared to the qPCR [35,42]. In this study, the specific primers and probe targeting APPV 5′ UTR were designed, and the qRT-PCR and cdRT-PCR for detection of APPV were successfully developed. The results showed that both methods could specifically test APPV, and no amplification signal was obtained from other swine viruses. The detection limit of the cdRT-PCR was 0.1 copies/μL, and that of the qRT-PCR was 10 copies/μL, demonstrating that the former method was 100 times more sensitive than the later method. Both methods showed excellent repeatability and reproducibility with CVs less than 5.27% (cdRT-PCR) and 0.90% (qRT-PCR), respectively. Both methods also showed excellent linearity between the initial number of templates and the Ct values/positive droplets, with R^2^ values of the standard curves exceeding 0.998. The application of both methods for the detection of 60 clinical samples showed a higher APPV positivity rate by the cdRT-PCR (25.00%, 15/60) compared to the qRT-PCR (23.33%, 14/60), and their coincidence rate was 98.33%. The sample with a Ct value of 36.12 by the qRT-PCR, which was considered as negative sample, was identified as positive sample with 1.23 copies/μL by the cdRT-PCR, indicating that the cdRT-PCR was more accurate and reliable for detection of the samples with a very low concentration. Some previous reports also established qRT-PCR or dRT-PCR for the detection of APPV [32,34,38]. The singleplex and multiplex qRT-PCR targeting the APPV NS3 and NS5B genes established by Yuan et al. [32] showed that the two assays had an R^2^ of ≥0.998 and E of ≥90%; the LOD of NS5B and NS3 genes of the two assays was 7.75 and 5.2 copies per reaction, respectively (in our study, the LOD was 250 copies per reaction); the intra- and inter-assay CVs were ≤1.46%. The multiplex qRT-PCR established by Liu et al. [34] showed that the R^2^ was ≥0.999 and E was ≥85%; the LOD was 2.52 × 10^1^ copies/μL (252 copies per reaction) for ASFV, CSFV, and APPV; the CVs of repeatability were ≤2%. The ddRT-PCR and qRT-PCR established by Deng et al. [38] showed that the R^2^ was 0.999 and 1, respectively; the LOD was 0.15 copies/μL and 9.2 × 10^1^ copies/μL, respectively; the CVs of repeatability were ≤5.96% for the ddRT-PCR. These assays demonstrated similar results to those of our study. Therefore, our results indicate that the established cdRT-PCR and qRT-PCR are specific and sensitive methods for the accurate detection of APPV.

APPV has been reported worldwide, causing certain economic damages to the swine industry. The APPV prevalence of serum samples collected from apparently healthy pigs was 2.3–17.5% in Europe and 5–11% in China [5]. The 369 serum samples from apparently healthy pigs in 28 German farms demonstrated 2.4% individual prevalence and about 10% prevalence at the farm level [43]. In litters with evidently healthy and CT-affected piglets, the fatality rate ranged from 17.2% to 24.6% in CT-piglets, and from 6.5% to 12.7% in healthy-piglets, while the fatality rate was up to 46.4% in litters comprising APPV-infected piglets affected by CT [28,44]. During the outbreak of CT, productivity dropped by about two weaned piglets per sow and year (about 10% drop) [20]. In four infected farms observed in Brazil in 2017, the CT rates were 16–100%, and the rates of splay legs were 6–55% in the CT-affected litters; the case fatality rate ranged from 0.5% to 30% [25]. Of the 1080 sera from Switzerland during 1986–2018, 9% (96/1080) were positive for APPV, varying from 0.3% in breeding farms in 2018 to 7–18% in fattening farms in 1986–2015 [45]. Yuan et al. [46] reported a 19.0% prevalence of APPV in a retrospective analysis of 1785 samples collected from 2016 to 2018 in the USA. Schwarz et al. reported that, in an infected pig farm in Austria, 80% of the litters were affected, and 70% of newborn piglets showed CT [20]. These results indicated that APPV is widely circulating in obviously healthy and CT-affected pigs worldwide, and its harmfulness cannot be ignored.

In China, many provinces have reported the discovery of APPV. Yuan et al. reported that a brief survey was performed in Guangdong province, southern China, in 2016, and most CT-affected pig farms had a 1–2% prevalence, with a positivity rate of 5.2% found in 135 collected serum samples from 10 farms [47]. A commercial pig farm in Guangdong province suffered an outbreak of CT in newborn piglets in 2016, with morbidity and mortality rates of about 2.67% and 60% in the CT-affected piglets [13]. The 440 sera and tissue samples from 27 farms in three provinces from June 2016 to January 2018 showed that 279 samples (63.4%) and 105 piglets (79.5%) were positive for APPV. The samples from Guangdong, Guangxi, and Anhui provinces demonstrated APPV positivity rates of 70.2% (179/255), 54.8% (63/115), and 52.9% (37/70), respectively [14]. The 165 sera from newborn piglets (all in their first week of life) from 21 different farms in southwest China from August 2017 to May 2018 showed that APPV was detected in 17 out of 39 sera (43.6%) from newborn piglets with CT on seven different farms [15]. The 83 samples from piglets with severe CT, which were obtained from 12 farms in Guangdong, Jiangxi, and Anhui provinces from May 2017 to March 2018, showed that 100% (12/12) of farms and 90.4% (75/83) of samples were APPV-positive, and 10% to 60% newborn piglets in different litters showed CT signs [31]. In this study, the 60 clinical samples from 15 pig farms in Guangxi province in 2020 and 2021 were tested using the established cdRT-PCR, and the positive rate was 25.00% (15/60), showing a high infection rate in the newborn piglets with congenital tremors. All results indicated that APPV is circulating in some pig herds in China. Therefore, more attention should be paid to the detection, surveillance, and epidemiological investigation of APPV in order to accurately determine the epidemic situation in the field.

APPV shows a high degree of genetic diversity [4,5,12,15,16,20,39,48]. To establish a sensitive and accurate PCR for testing this virus, the conserved regions of the viral genome are usually selected as the target genes for detection. In this study, the genomic sequences of different APPV strains were downloaded from NCBI GenBank (https://www.ncbi.nlm.nih.gov/nucleotide/ (accessed on 26 March 2023)), compared with DANMAN V6.0 software (http://dnaman.com, accessed on 26 March 2023), and the conserved region of 5′ UTR was selected to design the primers and probe (Table 1). After design, their sequences were further blasted in NCBI GenBank (https://blast.ncbi.nlm.nih.gov/Blast.cgi (accessed on 26 March 2023)) to ensure that they matched with most strains and maintained their specificity for APPV 5′ UTR. Of course, since the genetic variation of APPV strains continues to occur [16,17,39,48], new genetic variant strains are constantly emerging in the field. Therefore, the primers and probes should be updated and validated in a timely manner to ensure the accurate detection of newly emerging variant epidemic strains in clinical practice.

So far, it remains difficult to cultivate APPV, and no suitable cell line has been found to cultivate this virus in order to obtain high-titer pure virus for its identification [2,17,20,23,25,27]. In this study, we attempted to isolate and identify APPV, but failed. No animal experiment was conducted; thus, no sample from the experimental infected animals was obtained to validate the developed methods. However, APPV has widespread tissue tropism [20,25,26]. Therefore, the homogenized tissues from each pig (including the pigs suspected to be infected and healthy pigs) were used to detect APPV, aiming as best as possible to not miss any positive samples from different clinical forms of this disease. The 60 clinical tissue samples were analyzed using the two established methods, and their results indicated a coincidence rate of 98.33%, with similar positivity rates.

## 5. Conclusions

Specific primers and corresponding probe were designed targeting the 5′ UTR of APPV, the reaction conditions were optimized, and a cdRT-PCR and qRT-PCR were successfully developed and applied to test the clinical samples to evaluate the application of both methods. The results indicated that both methods were highly sensitive, strongly specific, and excellently reproducible; hence, they can be applied for the detection, surveillance, and epidemiological investigation of APPV to provide effective prevention and control measures. Each method has its own characteristics, which can be selected to test APPV according to the situations of different laboratories. The is the first reported cdRT-PCR for detection of APPV.

## Figures and Tables

**Figure 1 vetsci-10-00330-f001:**
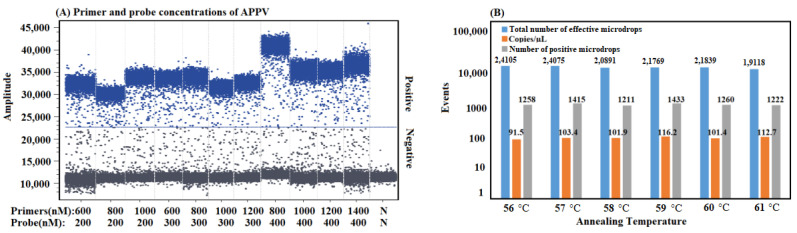
Optimization of the reaction conditions of the cdRT-PCR. The figures illustrate the amplification results of p-APPV plasmid at a final concentration of 10^3^ copies/μL with different primer/probe concentrations (**A**), and different annealing temperatures (**B**). N: negative control.

**Figure 2 vetsci-10-00330-f002:**
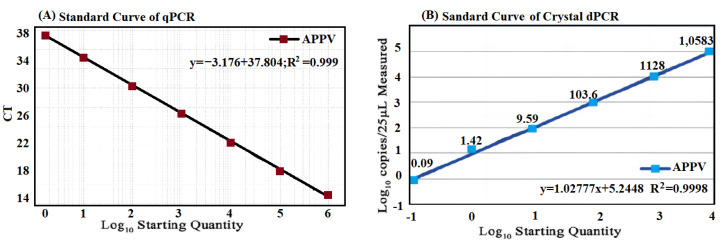
Formation of the standard curves. The final concentrations of p-APPV plasmid ranged from 10^6^ to 10^0^ copies/μL of the qRT-PCR (**A**), and from 10^4^ to 10^−1^ copies/μL of the cdRT-PCR (**B**).

**Figure 3 vetsci-10-00330-f003:**
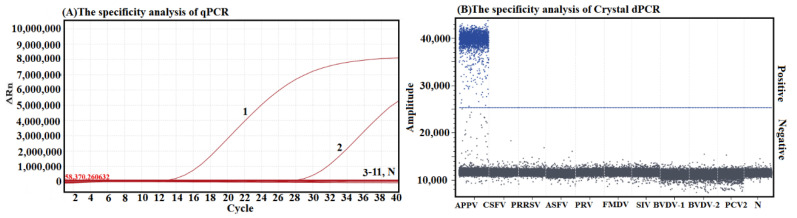
Specificity analysis. Figures show the results of the qRT-PCR (**A**) and the cdRT-PCR (**B**) using different viruses. In (**A**), 1–11: p-APPV, APPV, CSFV, PRRSV, ASFV, PRV, FMDV, SIV, BVDV-1, BVDV-2, and PCV2, respectively; N: negative control.

**Figure 4 vetsci-10-00330-f004:**
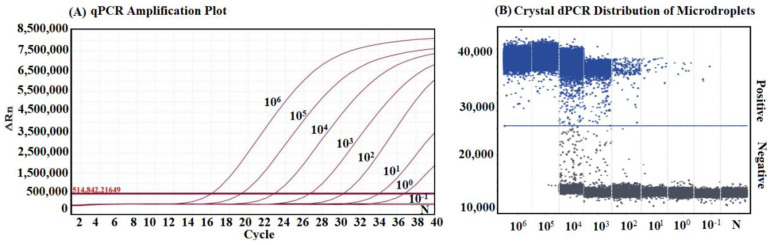
Sensitivity analysis. Figures show the dynamic curves of the qRT-PCR (**A**) and the droplet distributions of the cdRT-PCR (**B**) with different concentrations ranging from 10^6^ to 10^−1^ copies/μL (final concentrations). N: negative control.

**Table 1 vetsci-10-00330-t001:** The used primers and probe.

Primer/Probe	Sequence (5′→3′)	Product/bp
APPV-F	GGCGTGCCCAAAGAGAAAT	90
APPV-R	GGCACTCTATCAAGCAGTAAGGTCTA	
APPV-P	FAM-TCGGGTCCACCATGCCCCTTT-BHQ1	

**Table 2 vetsci-10-00330-t002:** The used reaction system.

	cdRT-PCR		qRT-PCR	
	Volume (μL)	Final Concentration (nM)	Volume (μL)	Final Concentration (nM)
qScript XLT One-Step RT-qPCR ToughMix (2×) (Quanta Bio., Gaithersburg, MD, USA)	12.5	1×	/	/
Fluorescein sodium salt (1000 nM) (Apexbio Bio., Beijing, China),	2.5	100	/	/
One Step RT-PCR Buffer (2×) (TaKaRa, Dalian, China)	/	/	12.5	1×
Ex Taq HS (5000 nM) (TaKaRa, Dalian, China)	/	/	0.5	100
Primer Script RT Enzyme Mix (5000 nM) (TaKaRa, Dalian, China)	/	/	0.5	100
APPV-F (25,000 nM)	0.8	800	0.4	400
APPV-R (25,000 nM)	0.8	800	0.4	400
APPV-P (25,000 nM)	0.4	400	0.3	300
Total nucleic acids	2.5	/	2.5	/
Distilled water	Up to 25	/	Up to 25	/

**Table 3 vetsci-10-00330-t003:** Evaluation of the repeatability and reproducibility.

Concentration	cdRT-PCR				qRT-PCR			
	Intra-Assay		Inter-Assay		Intra-Assay		Inter-Assay	
	$\bar{X}\pm \mathrm{SD}$	CV%	$\bar{X}\pm \mathrm{SD}$	CV%	$\bar{X}\pm \mathrm{SD}$	CV%	$\bar{X}\pm \mathrm{SD}$	CV%
10^3^	1114.67 ± 22.23	1.99	1121.22 ± 9.14	0.82	26.70 ± 0.24	0.90	26.64 ± 0.09	0.34
10^2^	109.07 ± 4.78	4.38	108.84 ± 0.63	0.58	30.20 ± 0.18	0.60	30.23 ± 0.24	0.79
10^1^	9.68 ± 0.51	5.27	9.50 ± 0.20	2.11	34.57 ± 0.25	0.72	34.42 ± 0.15	0.44

**Table 4 vetsci-10-00330-t004:** Comparison of the detection results by both methods.

		qRT-PCR			Coincidence Rate (%)	Kappa Value
		Positive	Negative	Total		
	Positive	14	1	15	98.33%	0.955
cdRT-PCR	Negative	0	45	45		
	Total	14	46	60		

## Data Availability

Not applicable.

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
