# Peer review of "Development of a Crystal Digital RT-PCR for the Detection of Atypical Porcine Pestivirus"

_vetsci, 2023, doi:10.3390/vetsci10050330_

Round 1

Reviewer 1 Report

In the manuscript titled “A Crystal Digital RT-PCR for the Rapid and Accurate Detection of Atypical Porcine Pestivirus”, the authors successfully established a crystal digital RT-PCR for detection of atypical porcine pestivirus (APPV), and used this method to test the suspicious clinical samples. The assays demonstrated high sensitivity, strong specificity, and excellent repeatability. My comments on the manuscript are as follows:

1. Line 236-239: Please add the advantages of the cdPCR here.

2. The syntax and grammar of the manuscript requires slight adjustment.  For example, “rapid and accurate”  in Line 1 should be “Rapid and Accurate”.

Author Response

The Cover Letter

April 16, 2023

Revision notes

We have made all efforts to meet the comments and suggestions provided by the editor and reviewers. We have revised our manuscript carefully according to the reviwers’s suggestions. Our responses are described in detail as follows.

Reviewer 1:

In the manuscript titled “A Crystal Digital RT-PCR for the Rapid and Accurate Detection of Atypical Porcine Pestivirus”, the authors successfully established a crystal digital RT-PCR for detection of atypical porcine pestivirus (APPV), and used this method to test the suspicious clinical samples. The assays demonstrated high sensitivity, strong specificity, and excellent repeatability. My comments on the manuscript are as follows:

  1. Line 236-239: Please add the advantages of the cdPCR here.

Response: The advantages of the cdPCR have been added. Please see Lines 259-262 in the revised manuscript.

  1. The syntax and grammar of the manuscript requires slight adjustment.  For example, “rapid and accurate”  in Line 1 should be “Rapid and Accurate”.

Response: We have checked and revised the manuscript carefully. Please see the revised manuscript.

Best regards,

Kaichuang Shi

Guangxi CADC, China

Reviewer 2 Report

I reviewed the manuscript entitled “Development of A Crystal Digital RT-PCR for the Detection of Atypical Porcine Pestivirus” In this study authors develop and validate a crystal digital RT-PCR for the detection of APPV.

Overall, considering the novelty of this development for the diagnostic of this disease, I think it is an interesting study. This are my suggestions to improve the quality of this manuscript.

1-    Title- Please clarify what is the title of this study. In this portal of the journal this study is entteled as Development of A Crystal Digital RT-PCR for the Detection of Atypical Porcine Pestivirus. However, in the document, it is referred as A Crystal digital RT-PCR for the rapid and accurate detection of atypical porcine pestivirus. In my opinion, the first one is more accurate for the nature of this study.

2-    Introduction. This section should be improved including more information regarding the pathogenesis of this disease. Include information about the main sources of virus during the infection and which samples may be considered for the diagnosis (oral rectal, nasal swabs, blood). Include more information regarding the genetic variability of this virus, conserved regions in the genome. It will help to support the value of primers and probe developed in this study to support the genetic variability of this virus in nature.

3-    Methods. Include more information regarding the clinical samples included in this study. Since the intention of this study was to develop and validate this method, why samples collected from negative animals were not included in the study. Discuss it as a limitation of this study.

4-    Discussion. This section should be drastically improved. Most of the information discussed in this section seems to fit better in the introduction section. Instead discuss, how your results correlate with previous results published using this technology. Based on the genetic variability of this virus, what is the expectancy regarding the performance of this test in the field. It is expected to cover all genetic variants? Discuss about other potential samples to be used for the diagnostic of this virus. Expose the main limitations of this study regarding the absence of samples included from experimental infected animals. Is this test accurate to detect different clinic forms of this disease?

Author Response

The Cover Letter

April 16, 2023

Revision notes

We have made all efforts to meet the comments and suggestions provided by the editor and reviewers. We have revised our manuscript carefully according to the reviwers’s suggestions. Our responses are described in detail as follows.

Reviewer 2

I reviewed the manuscript entitled “Development of A Crystal Digital RT-PCR for the Detection of Atypical Porcine Pestivirus” In this study authors develop and validate a crystal digital RT-PCR for the detection of APPV.

Overall, considering the novelty of this development for the diagnostic of this disease, I think it is an interesting study. This are my suggestions to improve the quality of this manuscript.

  1. Title- Please clarify what is the title of this study. In this portal of the journal this study is entitled as Development of A Crystal Digital RT-PCR for the Detection of Atypical Porcine Pestivirus. However, in the document, it is referred as A Crystal digital RT-PCR for the rapid and accurate detection of atypical porcine pestivirus. In my opinion, the first one is more accurate for the nature of this study.

ResponseWe agree the reviewer’s suggestion. The title has been revised as follows: Development of A Crystal Digital RT-PCR for the Detection of Atypical Porcine Pestivirus.

  1. This section should be improved including more information regarding the pathogenesis of this disease. Include information about the main sources of virus during the infection and which samples may be considered for the diagnosis (oral rectal, nasal swabs, blood). Include more information regarding the genetic variability of this virus, conserved regions in the genome. It will help to support the value of primers and probe developed in this study to support the genetic variability of this virus in nature.

Response: The information on APPV pathogenesis and the viral tissue tropism has been added in Lines 50-56 in the revised manuscript.

The information on the high degree of genetic diversity, and the conserved regions in the genome has been added in Lines 74-80 in the revised manuscript.

  1. Include more information regarding the clinical samples included in this study. Since the intention of this study was to develop and validate this method, why samples collected from negative animals were not included in the study. Discuss it as a limitation of this study.

Response: The information of the clinical samples has been added. The samples were also collected from negative animals, and tested by the developed methods, but we did not describe these negative samples in the manuscript. We discuss this limitation in the revised manuscript. Please see Lines 92-94 and Lines 350-359 in the revised manuscript.

  1. This section should be drastically improved. Most of the information discussed in this section seems to fit better in the introduction section. Instead discuss, how your results correlate with previous results published using this technology. Based on the genetic variability of this virus, what is the expectancy regarding the performance of this test in the field. It is expected to cover all genetic variants? Discuss about other potential samples to be used for the diagnostic of this virus. Expose the main limitations of this study regarding the absence of samples included from experimental infected animals. Is this test accurate to detect different clinic forms of this disease?

Response: we compared the results in this study with those of other previous reports. Please see Lines 277-290 in the revised manuscript.

        The information of the high genetic diversity of APPV was added in Lines 74-80 in the part of Introduction. In order to detect all the different variant strains of APPV, the genomic sequences of different APPV strains were downloaded from NCBI GenBank, compared, and the conserved region of 5' UTR was selected to design the specific primers and probe. After designing, the sequences of the primers and probe were further blasted in NCBI GenBank to ensure the sequences matching with most strains and maintain their specificity for APPV 5' UTR. We discuss this content in the part of Discussion. Please see Lines 336-349 in the revised manuscript.

        In this study, no animal experiment was conducted, so no sample from the experimental infected animals was obtained to validate the developed methods. However, APPV has widespread tissue tropism. Therefore, the homogenized tissues of liver, spleen, tonsil and lymph nodes from each pig (including the suspicious infected pigs and healthy pigs) were used to detect APPV, and try not to miss any positive sample from different clinical forms of this disease as much as possible. We discuss this content in the part of Discussion. Please see Lines 350-359 in the revised manuscript.

Best regards,

Kaichuang Shi

Guangxi CADC, China

Round 2

Reviewer 2 Report

I like to thank the authors for their responses to my questions. At this point, I don't have more concerns about this manuscript. 

Author Response

Thank you very much!